# Micheliolide Enhances Radiosensitivities of p53-Deficient Non-Small-Cell Lung Cancer via Promoting HIF-1α Degradation

**DOI:** 10.3390/ijms21093392

**Published:** 2020-05-11

**Authors:** Peizhong Kong, K.N. Yu, Miaomiao Yang, Waleed Abdelbagi Almahi, Lili Nie, Guodong Chen, Wei Han

**Affiliations:** 1Anhui Province Key Laboratory of Medical Physics and Technology/Center of Medical Physics and Technology, Hefei Institutes of Physical Sciences, Chinese Academy of Sciences, Hefei 230031, China; gckongpz@163.com (P.K.); yangmiaomiaom@163.com (M.Y.); wdsudanese@yahoo.com (W.A.A.); niell@cmpt.ac.cn (L.N.); 2Science Island Branch of Graduate School, University of Science and Technology of China, Hefei 230026, China; 3Department of Physics, City University of Hong Kong, Tat Chee Avenue, Kowloon Tong 999077, Hong Kong; peter.yu@cityu.edu.hk; 4State Key Laboratory in Marine Pollution, City University of Hong Kong, Tat Chee Avenue, Kowloon Tong 999077, Hong Kong; 5Collaborative Innovation Center of Radiation Medicine of Jiangsu Higher Education Institutions and School for Radiological and Interdisciplinary Sciences (RAD-X), Soochow University, Suzhou 215123, China

**Keywords:** micheliolide, radiosensitizer, hypoxia-inducible factor-1α, vascular endothelial growth factor, p53

## Abstract

Micheliolide (MCL) has shown promising anti-inflammatory and anti-tumor efficacy. However, whether and how MCL enhances the sensitivity of non-small-cell lung cancer (NSCLC) to radiotherapy are still unknown. In the present paper, we found that MCL exerted a tumor cell killing effect on NSCLC cells in a dose-dependent manner, and MCL strongly sensitized p53-deficient NSCLC cells, but not the cells with wild-type p53 to irradiation (IR). Meanwhile, MCL markedly inhibited the expression of hypoxia-inducible factor-1α (HIF-1α) after IR and hypoxic exposure in H1299 and Calu-1 cells rather than in H460 cells. Consistently, radiation- or hypoxia-induced expression of vascular endothelial growth factor (VEGF) was also significantly inhibited by MCL in H1299 and Calu-1 cells, but not in H460 cells. Therefore, inhibition of the HIF-1α pathway might, at least in part, contribute to the radiosensitizing effect of MCL. Further study showed that MCL could accelerate the degradation of HIF-1α through the ubiquitin-proteosome system. In addition, the transfection of wild-type p53 into p53-null cells (H1299) attenuated the effect of MCL on inhibiting HIF-1α expression. These results suggest MCL effectively sensitizes p53-deficient NSCLC cells to IR in a manner of inhibiting the HIF-1α pathway via promoting HIF-1α degradation, and p53 played a negative role in MCL-induced HIF-1α degradation.

## 1. Introduction

Lung cancer is the leading cause of cancer incidence and mortality worldwide, and non-small-cell lung cancer (NSCLC) is the most common histological subtype of lung cancers [1,2]. Radiotherapy is widely used, due to its advantages for NSCLC patients, especially in cases where the tumour is unresectable or the patient is inoperable [3]. However, radioresistance still remains a main obstacle limiting the efficacy of radiotherapy. Extensive efforts have been made to understand the mechanisms underlying radioresistance. The factors that are involved in the development of radioresistance include enhanced DNA damage repair, tumor metabolism alteration, cell cycle redistribution, as well as changes in the tumor microenvironment [4]. In particular, hypoxia, a general hallmark of tumor microenvironment, has been reported to be associated with radioresistance and hypoxia-inducible factor-1 (HIF-1) plays a major role in hypoxia-related radioresistance [5,6].

HIF-1 is a heterodimer that is composed of a regulatory HIF-1α subunit and a constitutively expressed HIF-1β subunit [7]. As an oxygen-dependent transcriptional factor, HIF-1 is increased and it promotes the transcription of genes associated with angiogenesis, cell survival, glucose metabolism, and invasion under hypoxic conditions in cancer cells. The change of HIF-1α protein level in response to normoxia/hypoxia mainly depends on the regulation of HIF-1α degradation. Under normoxic conditions, HIF-1α is hydroxylated and then ubiquitinated, which leads to a rapid degradation with a half-life of 5–8 min [8,9]. On the contrary, the degradation of HIF-1α could be inhibited by hypoxia, resulting in the accumulation of HIF-1α. Except hypoxia, oxidative stress and oncogenes also promote the activity of HIF-1 [10,11,12]. Previous researches have shown that radiation could increase the level of HIF-1α protein in a subset of radioresistant lung cancer cell lines [13]. The mechanism study revealed that the activation of phosphatidylinositol 3-kinase (PI3K)/Akt/mTOR increases de novo protein synthesis of HIF-1α and enhanced the interaction between heat shock protein 90 (Hsp90) and HIF-1α after irradiation (IR) [13].

Elevated HIF-1α protein expression promoted the secretion of vascular endothelial growth factor (VEGF), which mediated tumor-protective response to radiotherapy via vascular protection or inhibiting IR-induced apoptosis by upregulating anti-apoptosis protein, Bcl-2 [13,14,15]. In addition, hypoxia-induced HIF-1α activated the expression of glucose transporter-1 (GLUT-1), which conferred the enhanced tumor antioxidant capacity linking to radioresistance through initiating a glycolytic tumor metabolism [16]. Consistently, a number of clinical studies have also confirmed that HIF-1α overexpression has tight association with the poor prognosis to radiotherapy in various cancer types [17,18,19]. Therefore, HIF-1 is a target of sensitization for cancer radiotherapy, and many HIF-1α inhibitors, including small molecular compounds and small interfering RNA, have been developed and exhibited significantly radiation sensitizing effect [20,21].

Micheliolide (MCL), a natural guaianolide sesquiterpene lactone derivative of parthenolide (PTL) discovered in michelia compressa and michelia champaca plants, has exhibited promising therapeutic efficacy towards inflammation and multiple cancers [22,23]. MCL and PTL both belong to sesquiterpene lactone (SL), which has biological and pharmacological activities, owing to its specific moiety, α-methylene-γ-lactone, which can react with biological nucleophiles, such as cysteine sulfhydryl groups of target proteins [24]. The chemical structure of MCL is shown in Figure 1A. Nuclear factor kappa B (NF-κB) is well known as a reactive target of SL and its activity was found to be inhibited by SL via preventing the degradation of IκB-α and IκB-β [25]. Interestingly, PTL has been shown to radiosensitize prostate cancer cells via inhibiting the NF-κB pathway [26]. Furthermore, PTL selectively exhibits a radiosensitizing effect on prostate cancer cells, but not normal prostate epithelial cells by activating NADPH oxidase and it mediates intense oxidative stress [27]. However, PTL is unstable under both acidic and alkaline conditions, and this limits its medicinal applications [28]. MCL is more stable than PTL and it can selectively inhibit the growth of acute myelogenous leukemia stem and progenitor cells [29]. Moreover, the dimethylamino Michael adduct of MCL, DMAMCL, is highly promising for the treatment of glioma for crossing the blood-brain barrier, a formidable obstacle for drugs to exert a therapeutic effect in vivo, and it preferentially accumulates in the brain and effectively inhibits glioma cell growth [30]. DMAMCL was recently approved for clinical trials in Australia, owing to the superior oral bioavailability and enhanced therapeutic potential (Trial ID: ACTRN12616000228482). However, the radiosensitizing effect of MCL on NSCLC and the possible underlying mechanisms are still not known.

In the present study, we assessed the radiosensitizing effects of MCL on NSCLC. Our results indicated that MCL sensitized NSCLC, especially p53-deficient cell lines, to radiation under both normoxia and hypoxia via promoting the degradation of HIF-1α protein. Moreover, we found that p53 played a negative role in the degradation of HIF-1α that is induced by MCL. These results provide some hints that MCL can be used to sensitize NSCLC to radiotherapy.

## 2. Results

### 2.1. MCL Inhibits Cell Growth in NSCLC

We measured the viabilities of H1299 and Calu-1 cells at 24 h after exposure to various concentrations of MCL for 6 h in vitro to evaluate the killing effect of MCL on NSCLC. The cell viabilities of H1299 and Calu-1 cells treated with 5 and 10 μM MCL for 6 h were still higher than 90%, indicating that MCL induced modest cytotoxicity at concentrations less than 20 μM, as shown in Figure 1B. Significant inhibition of cell viability was observed when the cells were treated with relatively high concentrations of MCL (≥20 μM) for 6 h. The values of inhibitory concentration at 50% growth (IC50) of MCL for H1299 and Calu-1 cell lines were 27.97 and 33.83 μM, respectively. These results suggest that MCL exerts a cell killing effect in a dose-dependent manner.

### 2.2. MCL Sensitizes NSCLC to IR under Both Normoxia and Hypoxia

The cell viability of H1299 and Calu-1 cells were determined with CCK-8 after IR with or without MCL treatment to determine whether MCL can sensitize NSCLC to IR. The relative cell viability of H1299 cells decreased to 27.65 ± 1.80% after 4 Gy of IR with 20 μM MCL treatment, significantly lower than that with IR treatment alone (69.80 ± 4.84%) or MCL treatment alone (47.32 ± 6.01%), and the relative cell viability of Calu-1 cells also showed a similar trend. as shown in Figure 2A. Consistently, the enhanced killing effect of MCL was also observed after IR with 8 Gy (Figure 2A). The colony formation assay was further performed to test the radiosensitizing efficiency of MCL in H1299 and Calu-1 cells (Figure 2B). The survival fractions of MCL-pretreated H1299 and Calu-1 cells were significantly lower than their respective controls (no MCL treatment) after exposure to the same IR dose (2–6 Gy). Table 1 showed an increased sensitizer enhancement ratio for Dq (SERDq), 1.62 of H1299 and 1.69 of Calu-1, following MCL treatment.

In general, hypoxia is a tumor microenvironment condition that plays pivotal roles in tumor progression and resistance to radiotherapy [21]. Therefore, we next determine whether MCL exerts a radiosensitizing effect in NSCLC under hypoxic condition. After 4 Gy of radiation under hypoxia, the relative cell viability of H1299 cells decreased to 83.07 ± 5.85% (Figure 2C), higher than that under normoxia (69.80 ± 4.84%) (Figure 2A), and the relative cell viability of Calu-1 cells decreased to 84.86 ± 9.27%, which is also higher than that under normoxia (73.56 ± 6.14%) (Figure 2A). These results confirmed that NSCLC cells were more resistant to IR under hypoxia than normoxia. Furthermore, after treatment with 4 Gy of radiation plus 20 μM MCL under hypoxia, the relative cell viability of H1299 cells decreased to 21.84 ± 1.18%, significantly less than that with IR alone (83.07 ± 5.85%) and MCL treatment alone (36.41 ± 4.55%), and the relative cell viability of Calu-1 cells decreased to 34.60 ± 1.14%, also significantly less than that with IR alone (84.86 ± 9.27%) and MCL treatment alone (50.46 ± 2.02%). A similar trend was also observed after a higher dose of IR (8 Gy) under hypoxia (Figure 2C). These results indicate that MCL sensitizes H1299 and Calu-1 cells to radiation under hypoxia as well as normoxia. Colony formation assay was performed to further confirm the radiosensitizing effect of MCL in NSCLC under hypoxia. Significantly decreased survival fractions of MCL-treated cells were observed compared to controls (irradiated alone) in both H1299 and Calu-1 cells (Figure 2D). The results in Table 2 showed SERDq increased to 2.59 in H1299 and 1.82 in Calu-1 following MCL treatment under hypoxia. In addition, radiosensitization of MCL at a dose of less than 20 μM was determined with cell viability assay and colony formation assay. Indeed, pretreating H1299 and Calu-1 cells with 5 or 10 μM MCL sensitized these cells to IR under both normoxia and hypoxia (Appendix A). However, the radiosensitizing effects of MCL at 5 and 10 μM were weaker than that at 20 μM (Appendix A, Appendix A), which indicated that the radiosensitizing effect was affected by MCL in a concentration-dependent manner.

Taken together, the aforementioned results suggest MCL exerts a radiosensitizing effect in NSCLC under both normoxia and hypoxia. Moreover, MCL exhibits stronger radiosensitization under hypoxia than that under normoxia.

### 2.3. MCL Inhibits Radiation- and Hypoxia-Induced HIF-1α Expression in NSCLC

We first tested whether MCL enhanced radiosensitivity of NSCLC via regulating the NF-κB pathway to understand the mechanism of radiosensitization of MCL. IR activated the NF-κB pathway as demonstrated by increased phosphorylation of IκBα, decreased total IκBα, and accumulated nuclear localization of NF-κB p65, as shown in Appendix A. However, the activation of the NF-κB pathway caused by IR was not affected by pretreatment with MCL, suggesting that NF-κB pathway might not be involved in the radiosensitizing effect of MCL on NSCLC. Since the HIF-1α protein is an important negative factor for tumor radiotherapy [31], which is elevated after IR, we next tested whether MCL exerted radiosensitizing effect via inhibiting HIF-1α pathway. The expression of HIF-1α was induced higher at 6 h after IR and doses above 4 Gy did not induce further higher HIF-1α expression in H1299 and Calu-1 cells, as shown in Figure 3A. However, radiation-induced HIF-1α expression was effectively inhibited with pretreatment of MCL (20 μM) in H1299 and Calu-1 cells followed by 4 Gy IR (Figure 3B). Furthermore, the results of RT-PCR also revealed that the level of VEGF, a well-known downstream target of HIF-1α, significantly increased after IR, but this elevated VEGF was also effectively inhibited by MCL treatment (Figure 3C). Meanwhile, the VEGF level was correlated with the HIF-1α level for the indicated treatments, suggesting that the inhibition of HIF-1α further downregulated the transcription level of VEGF.

It is known that HIF-1α is also activated under hypoxia and then activates some downstream genes associated with radioresistance. Hence, we investigated whether MCL could suppress hypoxia-induced HIF-1α expression in NSCLC. We first focused on the dynamics of intracellular HIF-1α expression in H1299 and Calu-1 cells under hypoxia. The increase of HIF-1α protein, especially the posttranslational modification form with higher molecular weight, was observed within 1 h and then reached a peak level around 2 h under hypoxia (Figure 3D). We pretreated H1299 and Calu-1 cells with MCL for 4 h prior to hypoxic culture to evaluate the efficiency of MCL in suppressing hypoxia-induced HIF-1α expression. MCL pretreatment markedly suppressed hypoxia-induced HIF-1α expression in both H1299 and Calu-1 cells, as shown in Figure 3E. Consistently, hypoxia-induced VEGF mRNA expression was also significantly reduced following the downregulation of HIF-1α protein after MCL treatment (Figure 3F). We performed cell viability assay to evaluate the radiosensitizing effect of MCL on H1299 following HIF-1α knockdown to further confirm that the decrease of HIF-1α induced by MCL conferred radiosensitization. The cells transfected with siRNA of HIF-1α showed >90% downregulation of HIF-1α expression (Figure 3G). Results of cell viability assay showed that MCL pretreatment did not sensitize HIF-1α knockdown H1299 cells to IR (Figure 3H,I). However, consistent with a previous study [21], siRNA-mediated HIF-1α downregulation markedly sensitized H1299 cells to IR under both normoxic and hypoxic conditions (Figure 3H,I). These results indicated that MCL sensitizes NSCLC to radiation based on its negative role in HIF-1α regulation and targeting inhibition of HIF-1α is an effective approach to reduce radioresistance of NSCLC.

### 2.4. MCL Promotes the Degradation of HIF-1α

The possible mechanism that could account for decreased MCL-mediated HIF-1α expression in NSCLC was then studied. The level of HIF-1α protein was decreased after MCL treatment in a dose-dependent manner in both H1299 and Calu-1 cells. The most significant decrease in HIF-1α protein level was observed when the concentration of MCL was 20 μM (Figure 4A). HIF-1α mRNA expression was detected with RT-PCR after MCL (20 μM) treatment to verify the possibility that decrease of HIF-1α protein could be due to inhibited transcription of HIF-1α gene. However, no changes in HIF-1α mRNA expression were detected in both H1299 and Calu-1 cells after treated with or without MCL (Figure 4B), which suggested that the loss of HIF-1α protein in response to MCL treatment did not contribute to the transcriptional regulation. Considering that accelerated protein degradation could also cause the decrease of HIF-1α protein, we measured the turnover rate of HIF-1α protein with cycloheximide (CHX) chase in H1299 and Calu-1 cells pretreated with or without MCL. CHX (100 μg/mL) was used to block total cellular protein synthesis and chase was performed at 2, 5, and 10 min. The decline of HIF-1α protein amount in MCL pretreated cells was faster than that without MCL treatment during the same period, suggesting that MCL accelerated the degradation of HIF-1α protein, as shown in Figure 4C. Since HIF-1α protein was degraded mainly through the ubiquitin-proteasome pathway, we performed immunoprecipitation experiments to detect the level of HIF-1α protein after a proteasome inhibitor (MG132) treatment in MCL-treated H1299 and Calu-1 cells. The inhibition of proteasome with MG132 (20 μM) resulted in the formation of polyubiquitinated HIF-1α, and MCL-treated H1299 and Calu-1 cells led to more polyubiquitinated forms of HIF-1α when compared with cells without MCL treatment (Figure 4D). 

Taken together, these results indicate that MCL treatment can induce the decrease in HIF-1α protein via promoting its ubiquitin-dependent degradation. 

### 2.5. p53 Attenuates Radiosensitizing Effect of MCL

Interestingly, we found that the sensitizing effect of MCL on H460 (wild-type p53) cells was very weak. The survival fractions of MCL-treated H460 were slightly lower than the control cells after exposure to the same dose IR (2–6 Gy) under both normoxia and hypoxia, and SERDq (Table 3) showed an increase to 1.07 under normoxia and 1.21 under hypoxia in H460 cells, markedly lower than those of p53-null H1299 and Calu-1 cells, as shown in Figure 5A and B. Consistently, no significant decrease in HIF-1α protein level was observed after MCL treatment in H460 cells (Figure 5C). Furthermore, we assessed the inhibiting capacity of MCL on IR or hypoxia-induced HIF-1α expression in H460 cells. IR only induced slight expression of HIF-1α and administering of MCL (20 μM) failed to block the HIF-1α expression after IR (Figure 5D). Although HIF-1α expression was greatly induced by hypoxia in H460 cells, MCL treatment did not affect this process (Figure 5E). In addition, MCL also did not affect the induction of VEGF mRNA following IR or hypoxic treatment in H460 cells (Figure 5F). These results provide evidence that p53 might impair MCL-mediated radiosensitization via inhibiting MCL-induced HIF-1α downregulation.

We engineered stable p53-expressing (P53) as well as the control vector-expressing (VC) H1299 cells and colony formation assay was performed to further confirm the antagonistic action of p53 against radiosensitizing effect of MCL. IR significantly decreased the survival fraction of MCL-treated cells compared to control (irradiated alone) in VC H1299 cells but not in P53 H1299 cells under both normoxic and hypoxic conditions (Figure 5G,H). The survival curve parameters that are presented in Table 4 showed SERDq increased to 1.89 and 2.10 in VC H1299 cells under respective normoxia and hypoxia in the presence of MCL, whereas the SERDq were 1.04 and 1.30 in P53 H1299 cells, indicating the radiosensitizing effect of MCL was markedly inhibited by p53. Consistently, MCL pretreatment effectively inhibited radiation- and hypoxia-induced HIF-1α expression in control and VC H1299 cells, but the HIF-1α expression was not affected by MCL treatment in P53 H1299 cells (Figure 5I,J). 

Taken together, these findings indicate that p53 strongly impair the radiosensitizing effect of MCL on NSCLC via inhibiting MCL-mediated HIF-1α decline. 

## 3. Discussion

PTL, a compound with a structure similar to MCL, has been reported to have antitumor activity against NSCLC [32,33]. However, it is unclear whether MCL has antitumor activity against NSCLC cells. In present study, we found that MCL inhibited the growth of NSCLC cells (H1299 and Calu-1) in a concentration-dependent manner. This finding further expanded the antitumor killing spectrum of MCL. Similar to dimethylaminoparthenolide (DMAPT, a modified form of PTL), MCL also could sensitize NSCLC cells to IR in a concentration-dependent manner (Figure 2 and Appendix A). However, DMAPT sensitized NSCLC to IR by inhibiting NF-κB activation and blocking DSB repair [34], whereas the activation of the NF-κB pathway and DSBs repair induced by IR were not affected by MCL in the present study (Appendix A). In addition, previous studies suggested that cell cycle distribution and ROS production were involved in DMAPT-induced radiosensitization [35,36]. In contrast, MCL treatment did not affect the cell cycle distribution and ROS production after IR in H1299 cells (Appendix A). Thus, MCL and DMAPT are different in aspects of mechanisms of radiosensitization in NSCLC cells. Moreover, we observed that MCL exhibited higher radiosensitization under hypoxia when compared with normoxia, indicating that MCL also plays a negative regulatory role in hypoxia-induced radioresistance. Our finding suggests MCL combined with radiotherapy has a potential clinical application for attenuating the radioresistance under hypoxia, given that hypoxia is a typical characteristic of solid tumors [37].

Previous works suggested that HIF-1α is a critical determinant in response to radiotherapy [38,39]. The inhibition of HIF-1α expression has been regarded as a rational strategy to lower radioresistance of tumors owing to the induction of increased HIF-1α activity by IR and hypoxia [6,8,20]. In the present study, we also observed that HIF-1α expression was significantly elevated after IR or hypoxic exposure in both H1299 and Calu-1 cells. In particular, more induction of HIF-1α protein occurred following hypoxic exposure than IR. Consistently, higher radioresistance under hypoxia in both H1299 and Calu-1 cells confirmed the key role of HIF-1α in radioresistance. Moreover, our results showed that MCL treatment inhibited both IR- and hypoxia-induced HIF-1α expression in H1299 and Calu-1 cells. Knocking down HIF-1α distinctly attenuated the radiosensitization of H1299 cells by MCL, thus confirming that HIF-1α is a key target for radiosensitization by MCL. Additionally, the IR- and hypoxia-induced VEGF mRNA expression was also inhibited by MCL, reflecting the inhibition of HIF-1α /VEGF pathway. VEGF, one of the target genes of HIF-1α, was increased after exposure to IR or hypoxia, and inhibiting VEGF or VEGFR were effective strategies for improving the tumor radiosensitivity in clinical studies [40,41,42]. Therefore, we speculated inhibition of the HIF-1α/VEGF pathway might account for the radiosensitizing effect of MCL. This speculation was also supported by another study which suppression of the HIF1-α/VEGF pathway with NVP-BEZ235 or UO126 treatment resulted in the radiosensitization of endometrial cancer cells [43]. 

It is known that protein expression can be regulated at transcriptional and protein level. In our study, we observed that MCL did not affect the HIF-1α mRNA expression but decreased HIF-1α protein, indicating that MCL modulated the expression of HIF-1α at the protein level. The regulation of HIF-1α at the protein level is achieved through protein synthesis and degradation. The activation of phosphatidylinositol 3-kinase (PI3K) or mitogen-activated protein kinase (MAPK) pathway was reported to involve in HIF-1α synthesis [44,45]. In relation, HIF-1α degradation is regulated primarily via ubiquitin-proteosome system [12]. Moreover, the results of CHX-chase assay indicated that MCL impaired HIF-1α protein stability via promoting the degradation rate of the HIF-1α protein. These results were confirmed by our further investigation which showed that ubiquitination level of HIF-1α was higher in the presence of MCL when compared to that in the absence of MCL after MG132 treatment. Meanwhile, MCL-induced HIF-1α ubiquitination also suggested that MCL affected HIF-1α stability in the ubiquitin-proteasome pathway. Recent reports demonstrated that the degradation of HIF-1α was blocked by histone deacetylases (HDACs), especially HDAC1 and HDAC3, which could bind to oxygen-dependent degradation (ODD) domain of HIF-1α. In addition, HDAC1 regulates the reduction of the von Hippel-Lindau protein (pVHL), a ubiquitin ligase (E3) protein complex, which mediates the degradation of HIF-1α [46]. Most notably, PTL can specifically deplete cellular HDAC1 protein [47]. Thus, it is possible that MCL accelerates the process of ubiquitination and proteasomal degradation of HIF-1α via downregulating HDAC1. Certainly, further studies are needed in order to confirm this speculation. Interestingly, our results showed that no radiosensitization effect of MCL was observed in wild-type p53 cells. Consistently, MCL treatment failed to inhibit IR- and hypoxia-induced HIF-1α protein and VEGF mRNA expression in cells with wild-type p53. Our findings suggested that the radiosensitizing effect of MCL was attenuated by p53, owing to failure in HIF-1α inhibition. When considering the role of p53 in regulating radiosensitizing effect of MCL, the p53 status of individual NSCLC patient should be detected before using MCL as a radiation sensitizer during radiotherapy.

In summary, the present study shows that MCL sensitizes p53-null NSCLC cells to radiotherapy via inhibiting the HIF-1α expression and its downstream target VEGF. MCL inhibits HIF-1α expression partly through promoting the ubiquitin-dependent degradation of HIF-1α protein. However, the presence of p53 could attenuate the radiosensitizing effect of MCL via antagonizing the MCL-mediated HIF-1α decline (Figure 6). Our results provide support that MCL might be used as a sensitizer to improve the radiotherapy of NSCLC in the future. Certainly, the current results were obtained from experiments in vitro. When considering factors, such as bioavailability, distribution, and pharmacokinetics of MCL, might affect its clinical efficacy, the radiosensitizing effect of MCL on NSCLC will be further evaluated in vivo.

## 4. Materials and Methods 

### 4.1. Cell Culture and Reagents

The human NSCLC cell lines H1299 (p53-null), Calu-1(p53-null) and H460 (wild-type p53) were purchased from the Cell Bank of Type Culture Collection of Chinese Academy of Sciences (Shanghai, China). The H1299 and Calu-1 cell lines were cultured in RPMI 1640 medium (HyClone; GE Healthcare Life Sciences, Logan, UT, USA) and H460 was cultured in DMEM medium (HyClone), both being supplemented with 10% fetal bovine serum (FBS, HyClone), 100 μg/mL streptomycin (Gibco, Carlsbad, CA, USA) and 100 U/mL penicillin (Gibco). For normoxic cultures, the cells were maintained at 37 °C in a humidified incubator with 5% CO_2_ and 95% air. For hypoxic cultures, cells were maintained at 37 °C in a Whitley H35 Hypoxystation (Don Whitley Scientific, Shipley, UK) with 1% O_2_, 94% N_2_ and 5% CO_2_. All cell lines were free of mycoplasma. MCL (Wuhan ChemFaces Biochemical, Wuhan, China), protein synthesis inhibitor Cycloheximide (CHX, Sigma, St Louis, MO, USA), and proteasome inhibitor MG132 (Sigma) were dissolved in DMSO.

### 4.2. Irradiation

The cells were irradiated with a series of doses (0–8 Gy) with an X-ray irradiator (XHA600D, SHINVA, Zibo, China) at a dose rate of 0.189 Gy/min. 

### 4.3. Cell Viability Assay

Cells were treated with MCL (0~60 μM) for 6 h to assess the cell killing effect of MCL on NSCLC lines. Then the cell viability was measured after pretreatment with MCL (6 h before IR) pulse with IR to assess the radiosensitizing effect of MCL. For hypoxic exposure, four hours after MCL administering, the cells were moved into hypoxystation for additional 2 h before IR. The MCL-containing medium was replaced with normal culturing medium after IR and then cell viability was evaluated at 72 h after IR. Cell viability was measured with a Cell Counting Kit-8 (CCK-8, ApexBio, Houston, TX, USA). Two hundred microliters of CCK-8 solution was added to each well and incubated for 1 h at 37 °C. Absorbance at 450 nm was measured while using a Microplate Reader (Varioskan Flash, Thermo Fisher, Waltham, MA, USA). IC50 determination was performed using GraphPad Prism 7.0 software (GraphPad Prism Software, Inc., San Diego, CA, USA).

### 4.4. Colony Formation Assay

The cells were plated into 35 mm culture dishes at a density of 200 to 5000 cells/dish. After attaching on the surface of dishes, the cells were treated with MCL and radiation subsequently. The cells were treated with MCL for 6 h prior to radiation exposure. For hypoxic exposure, four hours after MCL administering, the cells were moved into hypoxystation for additional 2 h before IR. After IR, the MCL-containing medium was replaced with normal culturing medium and the cells were maintained under normoxic conditions. Eight to twelve days later, the cells were washed and then stained with 1% crystal violet, and the colonies containing ≥ 50 cells were counted. Plating efficiency (PE) was calculated by dividing the average number of colonies per dish by the number of cells seeded. The survival fraction (SF) was calculated by normalization to the PE of appropriate control groups. Survival curves were constructed with Origin 8.0 software (OriginLab, Northampton, MA, USA). The survival curve parameters, D0 and Dq, were calculated by fitting the data with the single-hit multi-target model [48].

### 4.5. Western Blot and Immunoprecipitation

Whole-cell protein was extracted with RIPA lysis buffer (Beyotime Biotechnology, Shanghai, China) and the protein concentration was determined with a BCA protein assay kit (Beyotime Biotechnology). The proteins were then separated with 6%–12% sodium dodecyl sulfate-polyacrylamide gel electrophoresis (SDS-PAGE) and then transferred to polyvinylidene fluoride membranes (Merck Millipore, Darmstadt, Germany). The membranes were blocked in 5% skim milk (BD/Difco, Sparks, MD, USA) for 1 h, and then incubated with different primary antibodies at 4 °C overnight. The primary antibodies utilized were: anti-HIF-1α (1:1000, Protein Tech Group, Wuhan, China), anti-ubiquitin (1:1000, Cell Signaling Technology, Beverly, MA, USA), anti-γ-H2AX (phospho S139) (1:1000, Abcam, Cambridge, UK), anti-IκBα (1:1000, Cell Signaling Technology), anti-phospho-IκBα (Ser32/36) (1:1000, Cell Signaling Technology), or anti-β-actin (1:1000, Protein Tech Group). After extensive washing with TBST, blots were incubated with IRDye-conjugated secondary antibodies (1:10000, Li-COR Biosciences, Lincoln, NE, USA) for 1 h at room temperature. Images of immunoreactive bands were captured with Odyssey CLx Infrared Imaging system (Li-COR Biosciences).

For immunoprecipitation (IP), 2 g of anti-HIF-1α (Abcam) incubated with 4 mg of cell lysate, followed by capturing with protein-A/G agarose (Beyotime Biotechnology). The beads were washed extensively and then suspended in SDS loading buffer for western blot analysis.

### 4.6. RT-PCR

RT-PCR was performed with One Step SYBR^®^ PrimeScriptTM RT-PCR Kits (Takara Bio, Dalian, China) on a Roche 480 Light Cycler (Roche, Basel, Switzerland). The primers used for PCR amplification are shown as follows: 5′-GAACGTCGAAAAGAAAAGTCTCG-3′, 5′-CCTTATCAAGATGCGAACTCACA-3′ (HIF-1α); 5′-AGGGCAGAATCATCACGAAGT-3′, 5′-AGGGTCTCGATTGGATGGCA-3′ (VEGF); and, 5′-CTGGGACGACATGGAGAAAA-3′, 5′-AAGGAAGGCTGGAAGAGTGC-3′ (ACTB). ACTB was used as a normalizing control and data analysis was performed as previously described [49], through calculating fold changes by the 2^−ΔΔ*C*t^ method.

### 4.7. Small Interfering RNA Transfection

The rransfection of cells with negative control RNA sequence (siNC) or HIF-1α small interfering RNA (siHIF-1α; GenePharma, Shanghai, China) were carried out with Lipofectamine 2000 Transfection Reagent (Invitrogen, Carlsbad, CA, USA) according to the manufacturer’s instructions. The target sequences of siHIF-1α are as follows: siHIF-1α-1: GGGTAAAGAACAAAACACA; siHIF-1α-2: AACTAACTGGACACAGTGTGT. Forty-eight hours after siNC and siHIF-1α transfection, the cells were used for the further experiment.

### 4.8. Retroviral Infection

The p53 overexpressed plasmid was generated by subcloning human full length TP53 cDNA (NM_000017.11) into the pCDH-CMV-MCS-EF1-Puro lentiviral plasmid. Lentiviruses were prepared through co-transfecting HEK293T cells with the P53 overexpressed plasmid and the packaging plasmids (psPAX2 and pMD2.G), as described previously [50]. The H1299 cells were infected with lentiviruses and the stable cell lines expressing p53 were selected for 10 days with puromycin (2 μg/mL) from 48 h after infection.

### 4.9. Statistical Analysis

All of the experiments were performed at least three times. The data were presented as mean ± SD. Differences were assessed using Student’s t test with SPSS v17.0 software (SPSS, Chicago, IL, USA), and cases with *p* < 0.05 was considered to be a statistically significant difference.

## Figures and Tables

**Figure 1 ijms-21-03392-f001:**
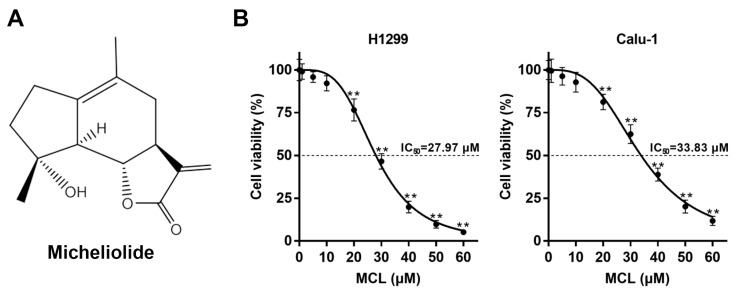
Chemical structure and cytotoxicity of micheliolide (MCL). (**A**) The chemical structure of MCL. (**B**) Dose-response curves of MCL for H1299 and Calu-1 cells. **, *p* < 0.01.

**Figure 2 ijms-21-03392-f002:**
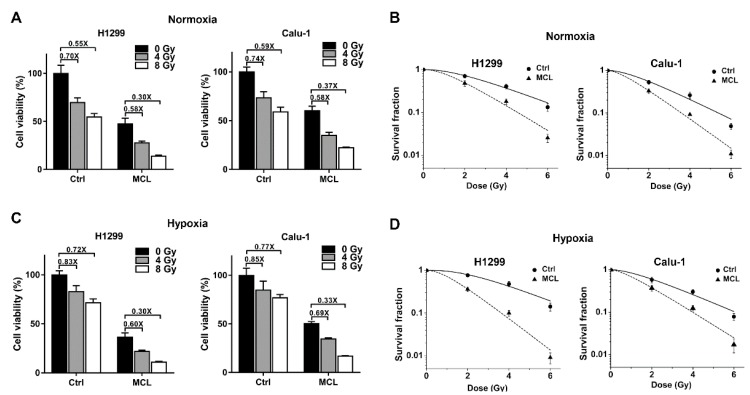
MCL sensitizes H1299 and Calu-1 cells to irradiation (IR). (**A**) The relative cell viability of H1299 and Calu-1 cells were evaluated at 72 h after IR with or without MCL (20 μM) pretreatment under normoxia. (**B**) The survival curves of H1299 and Calu-1 cells after IR with or without MCL pretreatment under normoxia. (**C**) The relative cell viability of H1299 and Calu-1 cells were evaluated at 72 h after IR with or without MCL (20 μM) pretreatment under hypoxia. (**D**) The survival curves of H1299 and Calu-1 cells after IR with or without MCL pretreatment under hypoxia.

**Figure 3 ijms-21-03392-f003:**
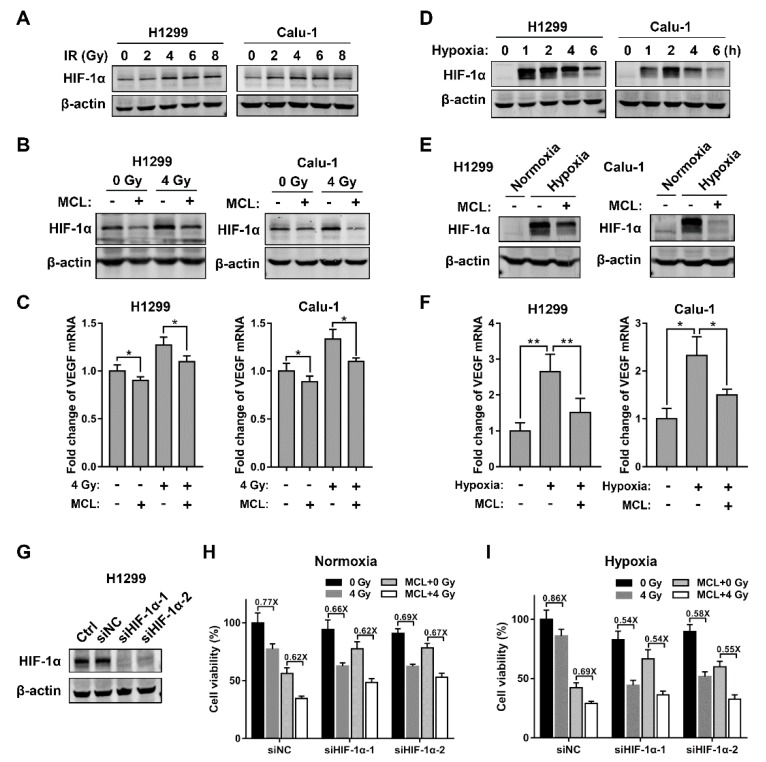
MCL inhibits radiation- and hypoxia-induced HIF-1α expression. (**A**) The expression of HIF-1α protein at 6 h after the indicated doses of IR in H1299 and Calu-1 cells. (**B**) The expression of HIF-1α protein at 6 h after the indicated doses of IR in H1299 and Calu-1 cells with or without pretreatment of 20 μM MCL. (**C**) The level of 5′-AGGGTCTCGATTGGATGGCA-3′ (VEGF) mRNA in indicated cells which were treated in the same way as in (**B**). *, *p* < 0.05. (**D**) The expression of HIF-1α protein at indicated time points after hypoxic exposure in H1299 and Calu-1 cells. (**E**) The expression of HIF-1α protein in indicated cells after exposing to hypoxia in the presence or absence of MCL. (**F**) The level of VEGF mRNA in indicated cells which were treated in the same way as in (**E**). *, *p* < 0.05; **, *p* < 0.01. (**G**) Determination of HIF-1α expression at 48 h after siHIF-1α transfection in H1299 cells. (**H**) The relative cell viability of siNC and siHIF-1α H1299 cells were evaluated at 72 h after IR with or without pretreatment of MCL (20 μM) under normoxia and hypoxia (**I**).

**Figure 4 ijms-21-03392-f004:**
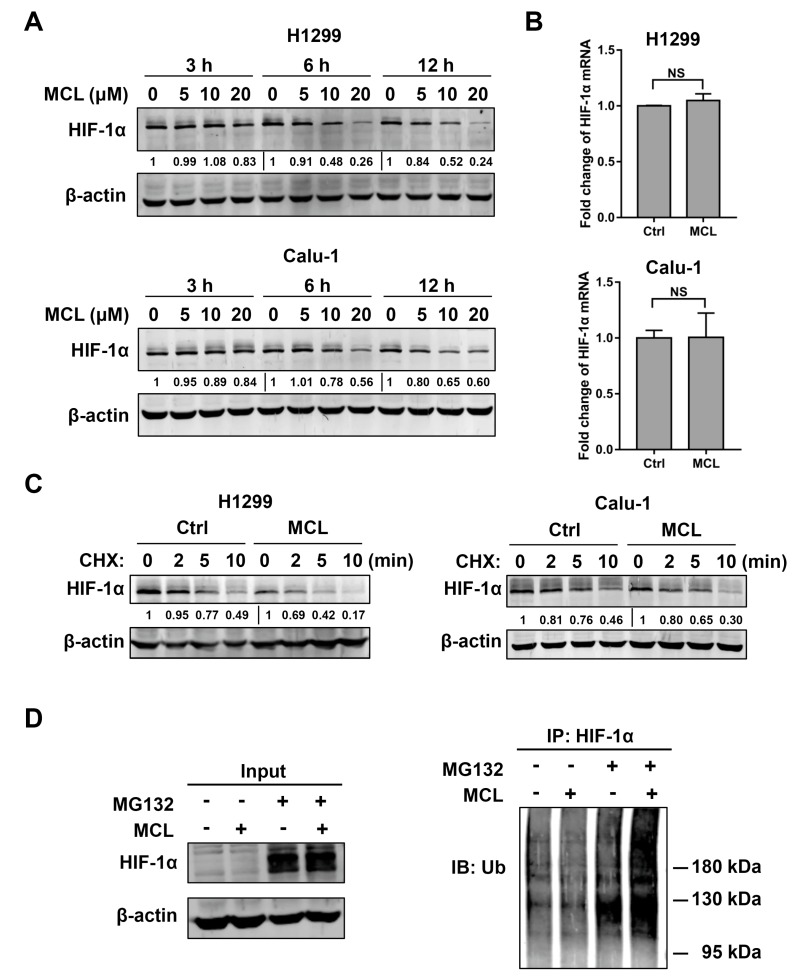
MCL promotes HIF-1α protein degradation in H1299 and Calu-1 cells. (**A**) The level of HIF-1α protein at indicated time points in H1299 and Calu-1 cells which were treated with various concentrations of MCL. (**B**) HIF-1α mRNA expression in H1299 and Calu-1 cells after the treatment of 20 μM MCL for 6 h. NS represents no significance. (**C**) The level of HIF-1α protein at indicated time points after cycloheximide (CHX) (100 μg/mL) treatment. Cells were pretreated with 20 μM MCL for 6 h, and then the MCL-containing medium was replaced with medium containing CHX. (**D**) The ubiquitination level of HIF-1α in H1299 cells which were treated with a proteasome inhibitor MG132 (20 μM) for 6 h in the presence or absence of 20 μM MCL.

**Figure 5 ijms-21-03392-f005:**
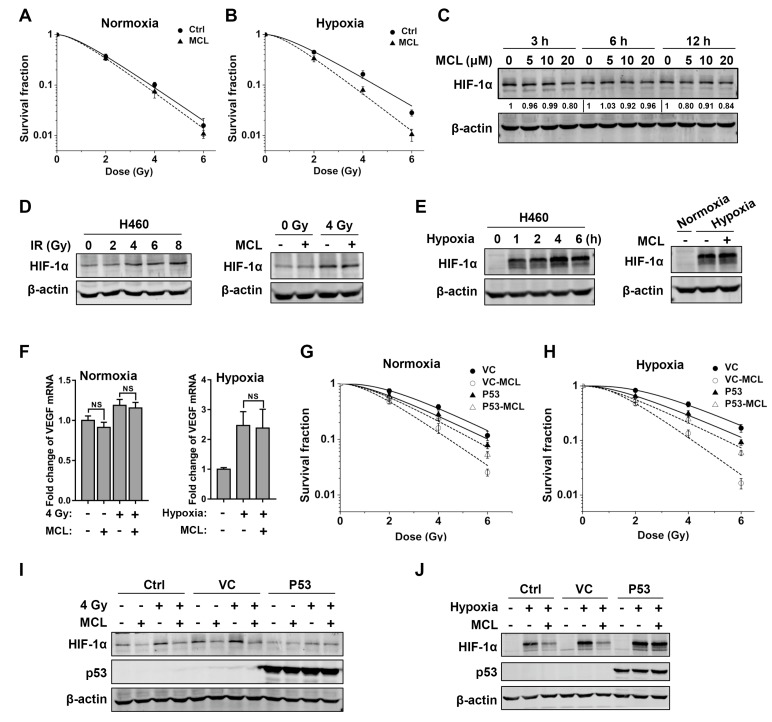
p53 attenuates the radiosensitizing effect of MCL by inhibiting MCL-mediated HIF-1α protein decline. (**A**) The survival curves of H460 cells after IR with or without MCL pretreatment under normoxia and hypoxia (**B**). (**C**) The level of HIF-1α protein at indicated time points in H460 cells after treatment with various concentrations of MCL. (**D**) The expression of HIF-1α protein in H460 cells treated with the indicated doses of IR in the presence or absence of 20 μM MCL. (**E**) The expression of HIF-1α protein at indicated time points in H460 cells after exposing to hypoxia in the presence or absence of 20 μM MCL. (**F**) The level of VEGF mRNA at 6 h post-IR or at 2 h under hypoxia in the presence or absence of 20 μM MCL in H460 cells. NS represents no significance. (**G**) The survival curves of VC and P53 H1299 cells after IR with or without MCL pretreatment under normoxia and hypoxia (**H**). (**I**)The effect of MCL pretreatment on the background and radiation-induced HIF-1α expression in indicated cell lines. (**J**) The effect of MCL pretreatment on the hypoxia-induced HIF-1α expression in indicated cell lines.

**Figure 6 ijms-21-03392-f006:**
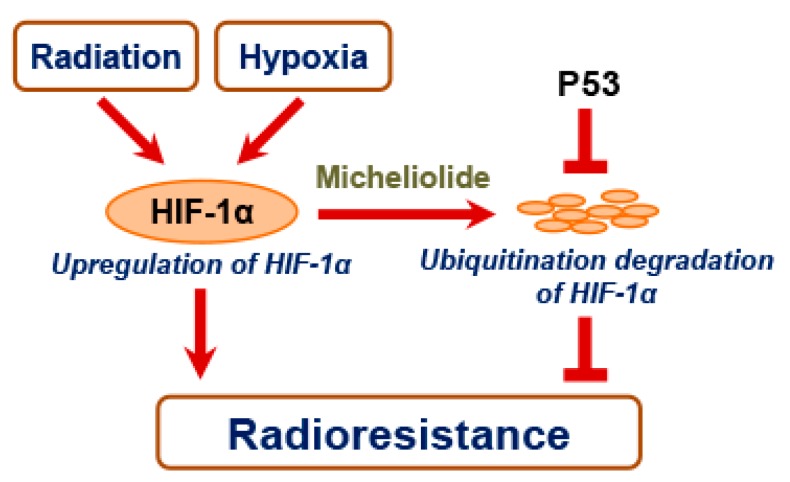
Schematic of MCL sensitizes p53-deficient NSCLC cells to IR via promoting HIF-1α degradation.

**Table 1 ijms-21-03392-t001:** The survival curve parameters of H1299 and Calu-1 cells after IR with pretreatment of MCL under normoxia.

	H1299			Calu-1		
	SF2	Dq	SERDq	SF2	Dq	SERDq
Ctrl	0.71 ± 0.06	1.95	-	0.53 ± 0.06	1.24	-
MCL	0.48 ± 0.07	1.20	1.62	0.33 ± 0.04	0.73	1.69

SF2, survival fraction at 2 Gy. Dq, quasithreshould dose. SERDq, sensitization enhancement ratio for Dq.

**Table 2 ijms-21-03392-t002:** The survival curve parameters of H1299 and Calu-1 cells after IR with pretreatment of MCL under hypoxia.

	H1299			Calu-1		
	SF2	Dq	SERDq	SF2	Dq	SERDq
Ctrl	0.77 ± 0.03	2.46	-	0.58 ± 0.09	1.33	-
MCL	0.36 ± 0.04	0.95	2.59	0.37 ± 0.06	0.73	1.82

SF2, survival fraction at 2 Gy. Dq, quasithreshould dose. SERDq, sensitization enhancement ratio for Dq.

**Table 3 ijms-21-03392-t003:** The survival curve parameters of H460 cells after IR with pretreatment of MCL under normoxia and hypoxia.

	Normoxia			Hypoxia		
	SF2	Dq	SERDq	SF2	Dq	SERDq
Ctrl	0.37 ± 0.04	0.91	-	0.45 ± 0.02	1.00	-
MCL	0.34 ± 0.03	0.85	1.07	0.33 ± 0.05	0.83	1.21

SF2, survival fraction at 2 Gy. Dq, quasithreshould dose. SERDq, sensitization enhancement ratio for Dq.

**Table 4 ijms-21-03392-t004:** The survival curve parameters of VC and P53 H1299 cells after IR with pretreatment of MCL under normoxia and hypoxia.

	Normoxia			Hypoxia		
	SF2	Dq	SERDq	SF2	Dq	SERDq
VC	0.74 ± 0.07	2.24	-	0.83 ± 0.03	2.78	-
VC-MCL	0.47 ± 0.03	1.18	1.89	0.47 ± 0.03	1.33	2.10
P53	0.60 ± 0.06	1.39	-	0.65 ± 0.04	1.67	-
P53-MCL	0.56 ± 0.07	1.33	1.04	0.55 ± 0.04	1.29	1.30

SF2, survival fraction at 2 Gy. Dq, quasithreshould dose. SERDq, sensitization enhancement ratio for Dq.

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
