# Peer review of "Micheliolide Enhances Radiosensitivities of p53-Deficient Non-Small-Cell Lung Cancer via Promoting HIF-1α Degradation"

_ijms, 2020, doi:10.3390/ijms21093392_

Round 1

Reviewer 1 Report

Authors have been addressed the comments raised by the reviewer and the quality of the manuscript has been improved. Unfortunately, authors did not conduct animal studies in the revised manuscript (please see the reviewer’s comment Q5). If possible, animal studies are recommended to perform, which will support their findings. At least, authors would properly discuss the limitations of the current results.

Author Response

We really thank the reviewer for helping us improve our manuscript, and we are sorry for not conducting animal studies. Due to the outbreak of COVID-19, nude mice are still in short supply, which cannot meet the needs of animal studies. According to the reviewer’s suggestion, we have discussed the limitations of the current results and added these into the last paragraph of Discussion section in the revised manuscript. The contents are as follows: Certainly, the current results were obtained from experiments in vitro. Considering factors such as bioavailability, distribution and pharmacokinetics of MCL might affect its clinical efficacy, the radiosensitizing effect of MCL on NSCLC will be further evaluated in vivo.

Reviewer 2 Report

Dear sirs,
according to my review requests, I really appreciate all the changes that were made by the authors. Thus, I encourage the publication of this paper in the present form.

Author Response

We are very grateful to the reviewer for his/her positive comments, and we really thank the reviewer for helping us improve our manuscript.

This manuscript is a resubmission of an earlier submission. The following is a list of the peer review reports and author responses from that submission.

Round 1

Reviewer 1 Report

Title: Micheliolide enhances radiosensitivities of p53-deficient non-small-cell lung cancer via promoting HIF-1α degaradation

The authors described the effect of micheliolide (MCL) on radiosensitivities of non-small-cell lung cancer cells. However, there are some drawbacks to be assessed before evaluation.

First of all, cytotoxicity of MCL on non-small-cell lung cancer cells, H1299 and Calu-1, at a dose of 20μM (Fig. 1B) can be thought to affect their results such as growth inhibition, radiosensitivities and so on. The authors should analyze their results using MCL at a dose of less than 20μM. MCL can cross the blood-brain barrier, a formidable obstacle for drugs to exert a therapeutic effect in vivo, and preferentially accumulates in the brain. The authors should mention this.

Reviewer 2 Report

Micheliolide (MCL), a natural guaianolide sesquiterpene lactone derivative of parthenolide, possessing anti-inflammatory and anti-tumor efficacy. In this study, author and colleagues analyzed the radiosensitizing effects of MCL on non-small-cell lung cancer (NSCLC). Their results showed that MCL sensitized p53-deficient NSCLC lines. This study further demonstrated that MCL enhanced the degradation of HIF-1α through the ubiquitin-proteosome system. Moreover, p53 played a negative role in MCL-induced HIF-1α degradation. Although this study is meaningful, it may be it may be necessary to address some key issues before accepting this manuscript.

1. In Fig. 1, authors claimed that “significant inhibition of cell viability was observed when the cells were treated with a high concentration of MCL (20 μM) for 6 h”. However, the results did not perform the statistical analysis. The statistical analysis for the experiment should be conducted. In addition, the LD50 of MCL on several NSCLC lines which were employed in the current study need to be performed.

2. Because nuclear factor kappa B (NF-κB) and oxidative stress played a central role in radiosensitizing other cancer cells, authors need to perform whether MCL enhanced radiosensitivity of NSCLC via regulating the NF-κB pathway and oxidative stress.

3. Fig. 4D showed that the formation of polyubiquitinated HIF-1α was unclear. Please provide a clear image of this figure.

4. DNA double-strand break (DSB) and cell cycle redistribution are involved in radioresistant of cancer cells. Since MCL exhibited a radiosensitizing effect of NSCLC in hypoxia, DSB analysis and cell cycle assay should be included in this study.

5. If possible, consider conducting animal studies in the revised work, which will greatly improve their findings and translate their results into clinical aspects.

6. In discussion section, several paragraphs that discussed their findings were similar to the description of the results. Please specifically discuss the main findings that related the others.

Reviewer 3 Report

Comments to Peizhong Kong et al.

This is an original and well-written paper, in line with journal’s aim and scope, where the main objective is to describe the micheliolide (MCL) as a promising anti-inflammatory and anti-tumor agent. Precisely the authors founded that MCL exerted tumor cell killing effect on NSCLC cells in a dose-dependent manner and that MCL strongly sensitized p53-deficient NSCLC cells but not the cells with wild-type p53 to irradiation, inhibiting the hypoxia-inducible factor-1α (HIF-1α) pathway.

The research was executed according to currently accepted standards, approach to statistical analysis of data is appropriate and the manuscript provides data that are likely to be of interest to journal readers. Summarizing, the paper is interesting to read, however, some minor corrections should be applied and listed below:

Please, use acronyms for sentences often used in the manuscript. The first time the acronym is used, it should be fully written out. IC50 determination, as well as intracellular and extracellular concentrations of MCL, were assessed by the authors? As described, MCL could cause different effects according to its concentration in the cell and these data could be interesting for readers. Please, specify the reasons regarding the time-point of MCL exposure (6h) chosen for the experimental procedures. Why the authors have selected only VEGF among HIF-1α targets? Please clarify. In order to understand the intracellular mechanisms induced by MCL exposure, especially in those networks strictly linked to HIF-1α activity, other gene- or protein-approaches could be used. If available, the authors should report some literature data regarding these topics. Finally, the authors should provide more updated bibliographic data. In order to facilitate the reader's comprehension, I strongly suggest moving figure 5k at the end of the discussion section, to mark the main conclusions of the work. Finally, it could be very interesting to use primary cell cultures obtained by surgical specimens in order to have more robust data to draw useful conclusions.
